# The Cannabis Plant as a Complex System: Interrelationships between Cannabinoid Compositions, Morphological, Physiological and Phenological Traits

**DOI:** 10.3390/plants12030493

**Published:** 2023-01-21

**Authors:** Erez Naim-Feil, Aaron C. Elkins, M. Michelle Malmberg, Doris Ram, Jonathan Tran, German C. Spangenberg, Simone J. Rochfort, Noel O. I. Cogan

**Affiliations:** 1Agriculture Victoria, AgriBio, Centre for AgriBioscience, Melbourne, VIC 3083, Australia; 2School of Applied Systems Biology, La Trobe University, Melbourne, VIC 3086, Australia

**Keywords:** breeding, plant productivity, medicinal cannabis, reproducibility, attribute associations

## Abstract

Maintaining specific and reproducible cannabinoid compositions (type and quantity) is essential for the production of cannabis-based remedies that are therapeutically effective. The current study investigates factors that determine the plant’s cannabinoid profile and examines interrelationships between plant features (growth rate, phenology and biomass), inflorescence morphology (size, shape and distribution) and cannabinoid content. An examination of differences in cannabinoid profile within genotypes revealed that across the cultivation facility, cannabinoids’ qualitative traits (ratios between cannabinoid quantities) remain fairly stable, while quantitative traits (the absolute amount of Δ^9^-tetrahydrocannabinol (THC), cannabidiol (CBD), cannabichromene (CBC), cannabigerol (CBG), Δ^9^-tetrahydrocannabivarin (THCV) and cannabidivarin (CBDV)) can significantly vary. The calculated broad-sense heritability values imply that cannabinoid composition will have a strong response to selection in comparison to the morphological and phenological traits of the plant and its inflorescences. Moreover, it is proposed that selection in favour of a vigorous growth rate, high-stature plants and wide inflorescences is expected to increase overall cannabinoid production. Finally, a range of physiological and phenological features was utilised for generating a successful model for the prediction of cannabinoid production. The holistic approach presented in the current study provides a better understanding of the interaction between the key features of the cannabis plant and facilitates the production of advanced plant-based medicinal substances.

## 1. Introduction

For millennia, *Cannabis sativa* L. (cannabis) has been extensively used by humankind as a multipurpose source for industrial [1], ritual [2] and pharmaceutical [3] applications. The production of these varied crops relies on the cultivation of distinct plant morphotypes that considerably differ in appearance and formation [4]. The current study focuses on “drug-type” cannabis plants that for generations have been cultivated to provide treatments for various medical conditions, including chronic pain [5,6], insomnia [7], epilepsy [8,9] and dermatological syndromes [10]. Although the medicinal effect of cannabis is a popular research topic, the development of scientifically based methods for cultivating and breeding advanced cannabis plants for the production of pharmacological substances is still in the early stages [11].

The medicinal properties of the cannabis plant are attributed to its secondary metabolites, specifically the terpenophenolic compounds classified as phytocannabinoids, which contain a large number of bioactive metabolites [12,13]. Phytocannabinoids (referred to hereinafter as cannabinoids) are synthesised in, secreted by and stored in trichomes, hair-like epidermal structures that can be found across most aerial parts of the cannabis plant but that appear in the highest abundancy over pistillate inflorescences [14,15,16,17,18,19]. Cannabinoids are key elements in the plant’s defence mechanism against biotic (e.g., insect predation) and abiotic (e.g., ultraviolet radiation and nutrient deficiency) stresses under which cannabis plants have evolved [20,21], which has generated a diversified array of cannabinoid compounds [19,22,23,24,25,26]. So far, 120 cannabinoids have been scientifically characterised, and these can be found in varying blends and ratios among different genotypes [27,28]. The most studied compounds, Δ^9^-tetrahydrocannabinol (THC) and cannabidiol (CBD), form a substantial proportion of the overall phytocannabinoid content and are considered “major cannabinoids”, while most other cannabinoids appear in trace quantities and are classified as “minor cannabinoids” [10,12,29,30].

To date, modern clinical studies that have investigated the medicinal properties of cannabis substances have focused primarily on the pharmacological potential of single cannabinoids to treat specific medical disorders [31,32], such as CBD for palliative care of Parkinson disease [33] or THC for the suppression of nausea in patients with cancer who are undergoing chemotherapy treatments [34]. However, over the years, a large number of publications have indicated that the therapeutic effect of whole-plant-based remedies is often superior to the effect obtained from drugs containing a single purified cannabinoid. This phenomenon is often referred to as “the entourage effect” and is attributed to synergic interactions between different chemical compounds that modulate the effect of the bioactive cannabinoids [35,36,37,38,39,40,41,42]. Thus, because the cannabinoid interplay plays a pivotal role in the pharmacological attributes of the remedies, understanding the array of factors that determine the plant’s chemotypic composition is crucial. Studies on the genetic control of cannabinoids’ biosynthesis have identified that the ratio between the major cannabinoids (THC/CBD) remains fairly stable regardless of environmental conditions or growth stage and can be considered as a *qualitative* trait that is regulated by a small number of genes. The B locus was previously proposed as the single defining factor affecting the variations in major cannabinoids [30]. However, recent studies have indicated that cannabinoid biosynthesis is more complex and affected by various factors, such as the number of gene copies [43] and the presence of paralogous genes that carry different levels of functionality [44]. In contrast to the biosynthesis of specific cannabinoids, the overall cannabinoid production is proposed to be a polygenic *quantitative* trait that can drastically vary across different environmental settings [45,46,47,48,49]. Although it is critical to understand the variation in the plant’s overall cannabinoid profile, research in this area is primarily limited to THC and CBD compounds, while the qualitative and quantitative traits of the minor cannabinoids have not yet been adequately explored [36,48]. The complex effect of genetic factors and environmental conditions on cannabinoid biosynthesis regulation poses a significant challenge for the pharmacological industry, as the production of cannabis-based phytomedicines requires a high level of consistency in the secondary metabolites’ qualitative (ratio between cannabinoids) and quantitative (the synthesised amount) traits [20,50,51,52,53].

To date, many cannabis studies have focused on the effect of specific cultivation factors on the plant’s morphology and cannabinoid composition. Saloner and Bernstein (2021) [54] examined the effect of nitrogen on cannabis productivity and reported that ample N supply can increase inflorescence biomass and reduce secondary metabolite concentration. Similarly, Shiponi and Bernstein (2021) [55] investigated the effect of phosphor applications on the performance of cannabis plants and indicated that above a certain threshold, P supply has a contrasting effect whereby an increase in inflorescences biomass is accompanied by a decrease in cannabinoid concentration. Light spectra and light intensity also play key roles in the determination of a plant’s morphology and the cannabinoid profile. Magagnini et al. (2018) [56] indicated that under different light sources, a given cannabis genotype might contain various cannabinoid profiles. Furthermore, agronomic techniques, such as the induction of water stress [57] and the manipulation of plant architecture [58], were also found to affect the production of secondary metabolites. These studies imply that by tightly governing the growth conditions and adopting strain-specific cultivation protocols, it is feasible to generate cannabis phenotypes containing the desired cannabinoid profiles. However, given that the cannabis plant is an intricate system combining physiological, phenological, morphological and agronomical attributes, we propose that a holistic understanding of the broader interplay between these factors may improve our understanding of the cannabis plant as a production unit. Previous research has identified associations between physiological, morphological and phenological parameters and inflorescence biomass [59,60]. The current study intends to build on these findings and examine whether these parameters are also associated with cannabinoid production and whether the deconvolution of these complex interplays can improve the prediction models of plant productivity.

Therefore, the objective of this study is to facilitate the production of advanced cultivars through the assessment of interrelationships between traits with high commercial value and to detect the key factors that regulate cannabinoid production and modulate their uniformity. To address this objective, we aim to investigate the interplays between the traits associated with the intact plant (e.g., yield, phenology, growth rate), inflorescence morphology and cannabinoid composition. In addition, cannabinoids’ broad-sense heritability indices are presented, and prediction equations for cannabinoid content that are based on the plant’s morphological and phenological parameters are exhibited, aiming to facilitate and accelerate the screening and selection process of superior cultivars.

## 2. Results

### 2.1. Phenotypic Diversity of Cannabinoid Compositions

To evaluate the diversity of cannabinoid production within the examined plant population, frequency distribution analysis was performed across the 119 tested genotypes, segregating them into subgroups according to their major cannabinoid profiles (high-THC and blended CBD:THC; Figure 1). Although most histograms associated with single cannabinoids are not characterised by a normal distribution (two out of seven, Figure 1A–G), the histograms of total synthesised cannabigerol (TS-CBG) and total cannabinoid concentration (TCC) are typified with a normal distribution pattern (Figure 1H and 1J, respectively). However, when the distribution pattern was examined separately for each of the major cannabinoid subgroups, a normal distribution was obtained across most histograms (Appendix A). Examinations of the differences in TCC between high-THC and blended THC:CBD subgroups indicate that genotypes from both groups span across the displayed range of the cannabinoid concentration chart (Figure 1J). However, for the total cannabinoids per plant (TCPP), the most prolific genotypes among the examined plant population are affiliated with the high-THC subgroup (Figure 1K). When examining the differences in the concentrations of cannabinoid variants, it is noticeable that the TS-CBG (Figure 1H, 25.37–143.72 mg/g) is in greater orders of magnitude than the total synthesised cannabigerovarin (TS-CBGV, Figure 1I, 0.06–4.08 mg/g), and in some genotypes, TS-CBG is up to 866-fold more concentrated than TS-CBGV (Appendix A). Moreover, when the distribution patterns of Δ^9^-tetrahydrocannabivarin (THCV) cannabidivarin (CBDV), and TS-CBGV (Figure 1B, 1D and 1I, respectively) are examined on the basis of the compound’s quantities, two distinct phenotypic groups can be identified within each chart (e.g., the subgroups’ range depicted in Figure 1I spans between 0–1.25 mg/g and 1.5–4.25 mg/g). However, no association between these groups and the plant’s major cannabinoid subgroups (high THC and blended THC:CBD) was observed.

### 2.2. Cannabinoids’ Broad-Sense Heritability

The estimated broad-sense heritability (H^2^) indices across the measured cannabinoid compounds range from 0.51 to 0.96 (Table 1). The H^2^ of TCC, which reflects a general heritability of cannabinoid biosynthesis, was evaluated at 0.62. Along with THCV and CBDV, the H^2^ values of THC and CBD were found to be the highest recorded entries, while the lowest H^2^ index for a cannabinoid compound was calculated for cannabinol (CBN, 0.51). Nevertheless, the relatively low H^2^ of CBN still exceeded all calculated heritabilities associated with plant phenology, the morphological attributes of the whole plant [excluding plant height (PH), H^2^ = 0.52] and the morphological characteristics of inflorescences. A comparison between the different categories of the recorded H^2^ (Table 1) identified that cannabinoid biosynthesis was characterised by the highest broad-sense heritability (mean H^2^ of 0.79), followed by plants’ phenology (H^2^ = 0.49), morphological traits of the intact plant (mean H^2^ of 0.4) and inflorescence morphological attributes, which were characterised by the lowest H^2^ (mean H^2^ of 0.24).

### 2.3. Trait Associations

Trait associations between cannabinoid compounds (TCC, TCPP, cannabichromene (CBC), CBD, CBDV, THC, THCV, CBN and cannabigerol (CBG)), the physiological and phenological traits of the intact plant (PH, inflorescence dry biomass (IDB) and days to maturation (DTM)) and inflorescence attributes (inflorescence size (IS), inflorescences number (IN) and inflorescence shape (ISH)) are presented alongside the performance of all examined genotypes (Figure 2 and Figure 3). The main two principal components (PCs) account for 54.4% of the variation across the examined parameters (32.2% and 22.2% for PC1 and PC2, respectively). Although plants’ vernacular classification into strain groups did not form distinctive clusters within the PCA matrix (symbol shapes, Figure 2), genotypic partitioning in accordance with the plant’s major cannabinoid profiles (high-THC and blended THC:CBD) did form clear group differentiations (symbol colour, Figure 2). Interestingly, while the two vectors’ aggregates of CBD, CBDV and CBC and THC, CBG, CBN and TCC were characterised by a negative association, no clear correlation was identified between these groups and vectors associated with plant productivity (IDB, TCPP and IN).

Along the cultivation period, the morphology of strain group 1 (SG1) genotypes (Figure 2, round red symbols) appeared different from those of drug-type morphotypes (e.g., relatively tall plants with a high number of internodes, accelerated growth rate during the reproductive phase, petite inflorescences and inferior harvest biomass). In order to statistically assess these differences across all parameters presented in Figure 2, a 95% confidence level ellipse was generated to evaluate the performance of SG1 genotypes in relation to all other examined genotypes (Figure 2, first and fourth quadrants of the Cartesian system). According to this assessment and in accordance with the distinctive position of SG1 genotypes over the PCA biplot, the overall profile of SG1 plants appears to diverge from the rest of the cannabis morphotypes evaluated in this trial. Therefore, to reliably reflect the properties of drug-type cannabis plants and to genuinely characterise associations between the physiological, morphological, phenological and chemotypic traits within these morphotypes, all SG1 genotypes (*n* = 9) were excluded from any further analysis in this study. In addition, owing to the distinct separation between high-THC and blended THC:CBD genotypes (Figure 2, black symbols and red symbols, respectively), the correlation analysis depicted in Figure 3 was performed separately for each of these groups.

Across both presented correlation matrices (high-THC and blended THC:CBD genotypes—Figure 3A and Figure 3B, respectively), no significant correlation was detected between IDB and IN to TCC. Moreover, in the analysis of both groups, no significant associations were found between a particular ISH and the biosynthesis of specific cannabinoids (except CBN). Furthermore, both strain sets demonstrated relatively similar correlation coefficients between TCPP to IDB (0.94 and 0.92—Figure 3A and Figure 3B, respectively), TCPP to PH (0.68 and 0.69—Figure 3A and Figure 3B, respectively) and TCPP to inflorescence width (Inf.W, 0.59 and 0.52—Figure 3A and Figure 3B, respectively). However, a significant correlation between vigorous growth rate during the reproductive phase (GR-R) and TCPP was found only for the high-THC strain group (Figure 3A, r = 0.46), while longer internodes (high Int.L) were found to be significantly correlated only with TCPP among the blended THC:CBD strain group (Figure 3B, r = 0.59). In addition, the correlation between TCC and TCPP appeared to be stronger for the blended THC:CBD strain group than for the high-THC strain group (0.61 and 0.48—Figure 3B and Figure 3A, respectively). Among the blended THC:CBD genotypes, plant precocity (low DTM) demonstrated a significant correlation with high CBC biosynthesis (Figure 3B, r = −0.39), and conversely, late maturity (high DTM) was found to be correlated with high CBC biosynthesis among the high-THC genotypes (Figure 3A, r = 0.46). Furthermore, a significant correlation between TS-CBG and TS-CBGV was demonstrated for the blended THC:CBD strain group only (Figure 3B, r = 0.49).

### 2.4. Variation in Cannabinoid Composition across Cultivars

Four vernacularly classified strain groups were selected to assess the variation in cannabinoid composition among genotypes affiliated with the same strain group (Figure 4). In all of the depicted charts, it is possible to identify a singular genotype that consistently underperformed relative to all other genotypes for most of the tested cannabinoids (relatively low mg/g). However, in some cases, when the overall cannabinoid production of this genotype is calculated per plant (TCPP, g/plant), its performance exceeded that of some genotypes in its group (e.g., orange and purple genotypes—Figure 4, SG2 and SG15, respectively). A comparison of TCC within strain groups indicated that the absolute cannabinoid concentration of some genotypes can be twice as high as in others (Figure 4, SG3—60.4 mg/g vs. 126.5 mg/g). Furthermore, across all the examined strain groups, CBG was the most variable cannabinoid, with an average ratio of 4.6 and a maximum ratio of 5.9 between its highest and lowest recorded quantities (Table 2 and Figure 4, strain group 7). With regard to the variation in the respective contents of the major cannabinoids, it was found that THC concentration and CBD concentration (mg/g) could vary up to 3.2 and 2.2 times, respectively, among the different genotypes (Table 2 and Figure 4, strain group 3). Moreover, the average concentration ratio between the highest and lowest THCV was estimated at 3.1 with a variation of up to 4.9 times between genotypes. Lastly, the mean ratio of TCPP was estimated at 2.2 with a variation of up to 3.3 times in the quantity of the overall cannabinoids between genotypes (Table 2, strain group 3).

### 2.5. Cannabinoid Variability within Genotypic Replicates

Figure 5 and Figure 6 demonstrate variations in cannabinoid concentration across plant replications. Although all plants within each radar chart (Figure 5) are genetically identical (clones) and were simultaneously cultivated at the same controlled environment (CE) facility, the fluctuations in major cannabinoid concentration (mg/g) across the examined genotypes are typified by a percentage difference of 25–43% for THC and 11–36% for CBD (Figure 5). The percentage difference in TCC levels within genotypes indicates that the overall cannabinoid concentration across clonal plants can vary by up to 39%, with absolute values ranging between 93.6 and 138.3 mg/g (Figure 5E).

Across both charts depicted in Figure 6, the greatest variation between the highest and lowest cannabinoid proportions across replicates was observed for the major cannabinoids (1.5% for THC and 3.4% for CBD—Figure 6A and 6B, respectively). Yet, the highest variation for some of the minor cannabinoids was typified by similar values (1.1% for CBG and 2% for CBDV—Figure 6A and 6B, respectively). However, when the absolute cannabinoid content, which corresponds with these percentages, is examined, it was found that the cannabinoid concentration (mg/g) between genotypic replicates can vary up to 2.6 times for major cannabinoids (12.8 mg/g to 33.2 mg/g for CBD, Appendix A) and up to 3.1 times for minor cannabinoids (0.14 mg/g to 0.44 mg/g for CBN, Appendix A).

In order to assess whether the ratios between cannabinoids remain constant across different replicates, the mean proportion and the standard deviation (σ) of each cannabinoid within each genotype were calculated (Table 3). Accordingly, for all cannabinoids, the calculated σ values are lower than 1%, except for CBD (σ = 1.41%) and THC (σ = 1.16%) in the THC:CBD blended strain (Table 3).

### 2.6. Prediction of Cannabinoid Production (TCPP) through Morphological, Physiological and Phenological Parameters

To predict a plant’s overall cannabinoid production (TCPP), multiple regression models were individually computed for the high-THC and blended THC:CBD strain groups, using morphological, physiological and phenological parameters of the intact plant and its inflorescences. These predictors include the PH on harvest day, Int.L, DTM and the average width of 10 inflorescences that were randomly selected from the plant’s largest main/apical inflorescences (Inf.W_10_). For both regression models, all assumptions regarding linearity (evaluated through partial regression plots and plots of studentised residuals against the predicted values), the independence of residuals (assessed via the Durbin–Watson statistic of 1.955 and 1.898 for high-THC and blended THC:CBD strain groups, respectively), homoscedasticity (inspected by plots of studentised residuals versus unstandardised predicted values), multicollinearity (tolerance values > 1) and normality (evaluated via Q-Q Plots) have been met. Each regression model significantly predicted TCPP for its targeted strain groups, and all predictors (except for Int.L for the blended group) added statistical significance to the model (Table 4), *p* < 0.05 (F(4, 69) = 34.98, *p* < 0.001, adjusted R^2^ = 0.65 for the high-THC strains and F(3, 32) = 20.86, *p* < 0.001, adjusted R^2^ = 0.63 for the blended THC:CBD strains).

According to the regression coefficients depicted in Table 4, Equations (1) and (2) were generated to predict the overall cannabinoid productivity (TCPP):

For high-THC genotypes,
(1)$${\mathrm{TCPP}}_{\frac{g}{plant}}=-21.64+0.13\times {\left(\mathrm{PH}\right)}_{\mathrm{cm}}-1.11\times {\left(\mathrm{Int}.L\right)}_{\mathrm{cm}}+0.19\times {\left(\mathrm{DTM}\right)}_{\mathrm{days}}+2.87\times {\left(\mathrm{Inf}.W_{10}\right)}_{\mathrm{cm}}$$

For blended THC:CBD genotypes,
(2)$${\mathrm{TCPP}}_{\frac{g}{plant}}=-13.68+0.07\times {\left(\mathrm{PH}\right)}_{\mathrm{cm}}+1.13\times {\left(\mathrm{Int}.L\right)}_{\mathrm{cm}}+1.77\times {\left(\mathrm{Inf}.W_{10}\right)}_{\mathrm{cm}}$$

In accordance with the coefficients in Equation (1) (high-THC genotypes), an increase of 1 cm in PH, a decrease of 1 cm in Int.L, prolonging plant maturity by 1 day and an increase of 1 cm in Inf.W_10_ are associated with an increase of 0.13 g, 1.11 g, 0.19 g and 2.87 g, respectively, in the overall cannabinoid production per plant. With regard to the coefficients in Equation (2) (blended THC:CBD genotypes), an increase of 1 cm in PH, 1 cm in Int.L and 1 cm in Inf.W_10_ are associated with an increase of 0.07 g, 1.13 g, 1.77 g, respectively, in the overall cannabinoid production per plant.

## 3. Discussion

### 3.1. Chemotypic Variation in the Examined Plant Population

Although it was ostensibly expected that prolonged and intensive selection pressure in favour of high psychotropic cannabinoid genetics (high THC content) had narrowed the overall chemotypic diversity among contemporary cultivars [20,61], the variation in cannabinoids detected within the study indicates a large chemotypic diversity remaining among different genotypes (Figure 1). However, because unadulterated cannabis landraces are scarce [22,43,61,62,63,64], the evaluation of the impact of recent breeding initiatives can be only subjective rather than absolute. Nevertheless, the current observation regarding broad chemotypic diversity is consistent with previous reports, suggesting that the recent decades’ intensive hybridisations [18,65] have generated a large number of cannabis genotypes characterised by a vast range of chemotypic profiles [66].

In general, histograms of parameters that are associated with the overall production of cannabinoids (e.g., TCC and TS-CBG) are often characterised by a unimodal continuous distribution and are frequently considered quantitative [29,43,47]. However, the parameters associated with the quantities of specific cannabinoids are often characterised by a bimodal pseudoqualitative distribution typified by discrete “peaks” corresponding to a specific allele’s composition. For example, the frequency distribution pattern of THCV (Figure 1B) implies that two major alleles are encoding for THCAS allozymes, each of which is characterised by different metabolic efficacies. However, because these histograms do not exhibit a purely discrete distribution pattern, the phenotypic variation around each of the bimodal peaks suggests that cannabinoid biosynthesis is still subjected to additional factors, which include (I) the availability of upstream precursors such as CBG and CBGV [67], (II) the absence or presence of enzymes competing on a common substrate for the synthesis of different end products, (III) the quantity of homologues gene duplication [68] and (IV) the diversity of paralogous genes coding for isoenzymes that carry varied levels of the fitness of activity [30,43,44,69].

### 3.2. Broad-Sense Heritability

The broad-sense heritability (H^2^) values presented in Table 1 indicate that under the current experimental settings, parameters within the cannabinoid category are the least susceptible to the effects of environmental factors (mean H^2^ of 0.79), while attributes associated with inflorescence morphology are highly affected by environmental conditions during cultivation (mean H^2^ of 0.24). These findings provide empirical support for previous studies that have suggested that environmental factors have a limited effect on the cannabinoid biosynthesis pathways [30,47,48]. However, the intermediate H^2^ value recorded for TCC (H^2^ = 0.62) suggests that environmental factors have an impact on the overall cannabinoid concentration. These findings correspond with previous observations that report that the absolute cannabinoid quantities may vary under different environmental conditions [30,46,48]. Among all cannabinoids, the lowest H^2^ index was obtained for CBN. Because CBN accumulation occurs as a result of THC degradation [70] in a process accelerated by exposure to environmental factors such as light, heat and oxygen [51,71], it was expected that in comparison to biosynthesised cannabinoids, CBN will be typified by a relatively low H^2^ index. Finally, while the relatively high H^2^ values depicted for most cannabinoids indicate a strong response to selection for these traits, future studies that explore the narrow-sense heritability (h^2^) of these traits will greatly benefit breeding programmes aiming to generate genotypes with a targeted chemotypic profile.

### 3.3. Strain Classifications and Trait Associations

Previously, broad inconsistencies were identified across the physiological and phenological traits of the intact plant [60] as well as across the inflorescence morphological attributes [59] among genotypes that are vernacularly associated with the same strain groups. The current study extends these findings and demonstrates that this inconsistency applies also to the plant’s cannabinoid profile (Figure 2 and Figure 4). This finding provides further support for the proposition that it is essential to adopt scientifically based methodologies for strains’ chemotypic classification in order to generate advanced pharmacological applications for cannabis products [72,73,74]. Additionally, our study identified significant interactions between traits associated with three parts of the cannabis plant: the physiology, morphology and phenology of the intact plant; its inflorescence morphology and its cannabinoid profile. Understanding the interplay between these traits is essential when evaluating the feasibility of generating genotypes carrying an array of desired attributes through selective breeding [75,76]. Accordingly, as no relationships were detected between plant productivity (IDB or TCPP) and the quantities of specific cannabinoids (Figure 3), it is plausible that breeding for high-yielding plants may not be limited to a certain chemotypic composition. Furthermore, as no association was observed between IDB and TCC (Figure 3), it is possible that high cannabinoid concentration in the harvested plant material does not necessarily come at a cost of low inflorescence biomass. Therefore, we propose that it will be feasible to simultaneously increase inflorescence productivity (g/plant) and cannabinoid concentration (mg/g) through selective breeding. Similarly, as no relationship was detected between a particular ISH and the quantities of specific cannabinoids (excluding CBN), it is plausible that any favourable form of inflorescence could carry a desirable cannabinoid profile (Figure 3). These observations support Woods et al. (2021) [77], suggesting that the loci of agronomic and biochemical traits are predominantly independent of each other. Regarding morphological parameters and cannabinoid productivity, our previous prediction equation for IDB showed the efficacy of Inf.W measurements in forecasting inflorescences yield [59], and building on this data, the current findings uncovered a positive relationship also between Inf.W and TCPP (Figure 3). Moreover, the positive association between TCPP and PH indicates that regardless of the genotypic partitioning based on the main cannabinoid profiles, high-stature plants are more productive (Figure 3). Finally, the positive association between DTM and CBC (in high-THC strains) implies that late maturation is associated with high CBC content (Figure 3). These findings correspond with the work of de Meijer et al. (2009) [78], demonstrating that CBC synthesis is more intensive during the juvenile phase and that, therefore, prolonged vegetative growth (high DTM) is expected to be correlated with high CBC content. However, it is notable that this was not observed in the blended THC:CBD strains group, wherein there was a negative association between late maturation (high DTM) and high CBC content. Because there are limited studies exploring this relationship, further investigations are required to clarify the interaction between precocity and CBC.

### 3.4. Cannabinoid Variation between and within Strain Groups

The pharmacological activity of cannabis drugs is not necessarily attributable to a single cannabinoid but instead to the synergism or “the entourage effect” of several compounds [36,37,38]. Hence, it is expected that plants carrying different chemotypic profiles might present with dissimilar therapeutic effects [74]. The large variation in cannabinoid quantity and quality (amount and ratio) observed within vernacularly classified strain groups (Figure 4 and Table 2) provides a reasonable explanation for the inconsistencies and lack of reproducibility of cannabis products [79]. Accordingly, in order to advance the development of modernised cannabis remedies, it is essential to adopt better classification methodologies to distinguish between strains and their therapeutic activities [51,53,72,80].

Although interstrain cannabinoid variation and its medicinal ramifications have been extensively discussed in the literature [20,53], research that focuses on differences in cannabinoid content across genotypic replicates remains limited. Previous studies have proposed that plant maturity [48,61,81], as well as variation in cannabinoid content during the diurnal cycle [51], may affect the harvested material’s chemotypic profile. The chemotypic variation presented suggests that plant maturity impacts CBN content but does not necessarily affect the other cannabinoids (Appendix A, columns 15–28, which were harvested in later stages of the season). However, there were still relatively large differences in the cannabinoid quantities found between replicates within the study (Figure 5), which indicates that although the study was conducted within a highly regulated CE facility, there was still an effect on the absolute quantities of the synthesised cannabinoids (Appendix A). Thus, to minimise environmental variations even when cultivation takes place in a CE facility, it is suggested to adopt cultivation practices that facilitate the uniform dispersal of environmental parameters, such as by shaping the desired plant architecture via pruning to allow better regulation of the microenvironment near the plant’s shoot [58] or by implementing systems providing a homogeneous spread of the light spectrum [82].

Alongside the variability in cannabinoid concentrations, the current study demonstrated that the ratios between cannabinoid absolute quantities (THC, CBD, CBC, CBG, THCV and CBDV) remain stable between plant replicates (Figure 6 and Table 3). Although this phenomenon has been extensively discussed in the literature, it is often limited solely to the ratios between THC and CBD [22,46]. As the THC:CBD ratio remains constant under different environmental conditions, several studies have proposed utilising it as a tool for chemovar classification [48]. However, limiting this classification to only THC and CBD has several drawbacks. These include classification errors that may occur when classifying genotypes that carry similar ratios of major cannabinoids (THC and CBD) as the same chemovar, regardless of variation in minor cannabinoid composition, or the inability to characterise strains lacking in CBD or THC. To overcome these issues, we propose establishing a chemovar characterisation method that is based on the relative proportions of six biosynthesised cannabinoids (CBD, THC, CBG, CBC, THCV and CBDV). A system that is based on this more extensive profile (rather than a purely CBD-to-THC ratio) will establish a more accurate, reliable and comprehensive classification that can be used for strain identification and will also act to protect breeders’ intellectual property. Further studies are needed to verify that this method will remain reliable under various cultivation environments.

### 3.5. Prediction Equation

As screening and plant selection, through phenotyping and chemotyping, are often costly, time-consuming and labour-intensive, a better solution to enable the rapid evaluation of large plant quantities is needed. In most cases, plant productivity, measured as inflorescence yield or oil content, is the typical metric that cannabis breeders and growers apply [52,83]. The prediction equations presented in this study aim to support the selection of high-yielding plants by using a simple and rapid method based on morphological and phenological traits to substantially accelerate and simplify the screening process. The application of the prediction equation, based on parameters that can be measured without advanced scientific equipment (such as UHPLC), will deliver reductions in time and economic costs to identify elite genotypes. In the current study, the bifurcation of the examined plant population into high-THC and blended THC:CBD groups generated a specific equation for forecasting cannabinoid production for each of the cohorts. This reflects that the traits that generate the predictions differ between the two groups. Previously, we demonstrated that IDB can be predicted solely by measuring the average width of 10 apical inflorescences [59]. Findings from the current study provide an exciting advancement for the prediction of cannabinoid production, whereby, it is the first model which demonstrates that an array of morphological and phenological parameters can be used for the prediction of TCPP. The morphological predictors of TCPP indicate that plant architecture potentially affects cannabinoid content, as taller plants that have a greater number of internodes generate more resin-containing inflorescences. The specific trait of Inf.W_10_ has the greatest effect on TCPP production; potentially, wider inflorescences that contain more florets will produce more glandular trichomes and thereby increase the overall resin content. These results suggest that combining data obtained from both the IDB and TCPP prediction equations can provide a powerful, easy-to-perform and comprehensive yield forecast for individual plants. However, as the phenotypic performances are subjected to the effects of environmental conditions, further studies are required to validate the accuracy of these prediction equations under various field conditions.

## 4. Materials and Methods

The work conducted in the current study was subjected to the Medicinal Cannabis Research Licence (RL011/18) and Permits (RL01118P6 and RLO1118P3) issued by the Australian Department of Health (DoH), Office of Drug Control (ODC). Descriptions of the methodologies undertaken to assess morphological, phenological and physiological attributes are briefly discussed below [59,60].

### 4.1. Plant Material, Experimental Design and Growth Conditions

The examined plant material in the current trial comprises 119 genotypes that were obtained from commercial medicinal cannabis companies (97 genotypes) or developed as hybrid lines by Agriculture Victoria Research (22 genotypes). Each of the tested genotypes was germinated from a genetically unique seed and formed a single stock (mother) plant. Mother plants were maintained in a vegetative phase and utilised to generate 10 cuttings (10.5 ± 0.5 cm) in coconut coir propagation plugs (Jiffy-7C, ø5 cm, Zwijndrecht, the Netherlands) using hormone solution (Growth Technologies, Clonex, 3 g/L IBA gel, Perth, Australia) to invigorate rooting. Seedlings establishment took place in a CE chamber under a relative humidity (RH) of 55%, a temperature of 24/18 °C (day/night) and a long daylight regime (16 h) provided by fluorescent light systems (Philips TL-D Reflex 58 W/840, Amsterdam, the Netherlands), delivering a PPFD of 360 (μmol × m^−2^ × s^−1^, measured 35 cm below the light source). Next, 30 days later, 6–7 plants of each genotype, similar in size and growth vigour, were selected and transplanted for further establishment in larger coconut coir propagation plugs (Jiffypots, ø8 cm, Zwijndrecht, The Netherlands). The trial replication quantity was determined through a preliminary study where 4–5 clonal replications were found to be adequate to calculate the genuine genotypic value for each trait despite differences in phenotypic performance arising from variations in environmental factors within the CE facility. Thus, on planting day, 4–5 uniform plants of each genotype were selected and planted into coconut coir grow slabs (100 × 16 × 10 cm, Cazna grow slabs, Sydney, Australia) that were placed on drainage trays situated upon a rolling-benches system. Planting spaces were set to 20 and 40 cm within and between rows, respectively, to form an overall density of 4.3 plants × m^−2^. Plants grew vegetatively under an 18 h daylight regime for 42 days before the reproductive phase was initiated by shortening the daylight cycle to 12 h. Across the CE facility, fixed high-pressure sodium lights (Philips, MASTER GreenPower Xtra 1000 W EL/5 X 6CT, Amsterdam, the Netherlands) were used to deliver a light intensity (PPF) of 2150 μmol × s^−1^. During the entire cultivation period, the temperature was set to 20/17 °C (day/night), and the RH was maintained at 60%. Fertigation was applied through a controlled drip irrigation system (Jain Octa-BubblerTM, 7.5 L/h, Fresno, CA, USA), provided 1% of A&B nutrient solution (THC, coco A + B, Melbourne, Australia) with an EC of 2.1 dS/m and a pH of 6–6.1. Although no pests were identified during the cultivation period, for pest management, beneficials (arthropods) were frequently dispersed. To prevent overshading or breaking, wooden stakes were attached to tilted plants where needed.

### 4.2. Determination of Morphological and Phenological Parameters

In the current trial, 532 individual plants were cultivated, and a comprehensive assessment of each plant’s phenological and morphological traits, inflorescence characteristics and chemotypic profiles was performed.

Through the cultivation period, plant height was measured weekly, and the recorded data were used to calculate growth rates during the vegetative (GR-V) and the reproductive (GR-R) growing phases. To evaluate plant phenology, the period between short daylight induction and the appearance of amber-shaded stigmas on 3 independent inflorescences was measured and defined as DTM. Plant harvest was performed selectively by frequently monitoring changes in stigma colour within each plant and carried out when ~70% of the overall plant’s stigmas turned brown. On harvest day, PH was recorded and the average Int.L was calculated as the quotient between the number of internodes forming the leading shoot and the PH. For each plant, vegetative (stem, leaves) and reproductive (inflorescences) substances were manually separated, and leaf trimmers (Growlush bowl trimmer, 19″, Melbourne, Australia) were used to refine inflorescence material by further removing vegetative organs such as leaves and petioles. Thereafter, all the processed inflorescences of each plant were distributed over a scaled surface and captured in an image taken perpendicularly above the surface centre by a dual-pixel camera (Samsung Galaxy S8, Seoul, Republic of Korea). All images were then processed and edited via GIMP (GIMP Development Team, 2.10.12, 2019), and in-house custom-made C++ software (C++17 ISO standard compliant) was utilised to perform image analysis, which provided empirical evaluations for an array of morphological traits characterising each inflorescence within each plant. In total, through the image analysis process, the data of 127,000 inflorescences were extracted, and parameters such as the IS (cm^2^), IN and the total inflorescences coverage (TIC, cm^2^) in each plant were recorded. Furthermore, inflorescence length (Inf.L) and Inf.W were evaluated via the best-fit ellipse that can enclose the object boundaries while factoring in the shape’s plasticity and variability. The ratio between Inf.W and Inf.L was computed to assess the ISH in a numerical manner, where values nearing 1 and 0 typify rounded and elongated inflorescences, respectively. Following processing and photographing, the inflorescences of each plant were placed in a drying chamber (25 °C, 20% humidity) for a minimum period of 14 days and transferred to a freeze dryer (VirTis, GPFD, Gardiner, NY, USA) for complete dehydration before the total IDB was recorded.

### 4.3. Cannabinoid Quantification

For each plant, all freeze-dried inflorescences were roughly ground to form a homogeneous mixture that was sampled for the chemotypic analysis. Sample extraction was performed as detailed in [84,85]. In short, 10 ± 0.2 mg of ground inflorescences was weighed into a 2 mL microtube and extracted with 1 mL of 80% methanol (*v*/*v*). A 100 µL aliquot of the supernatant was diluted 10-fold to 1 mL in a 2 mL LC-MS vial for analysis. The remaining supernatant was transferred to a separate 2 mL LC-MS vial for analysis of minor cannabinoids if necessary. All cannabinoid standards were purchased from Novachem Pty Ltd. (Heidelberg West, Australia) as the distributor for Cerilliant Corporation (Round Rock, TX, USA). Standards for CBDA, CBD, CBN, THC, CBC and CBCA were prepared as described in [85]. Standards for CBDV, CBDVA, CBNA, CBCA, THCV, THCVA, d8-THC, CBG, CBGA and CBL were prepared by diluting neat standards 10-fold to 100 µg/mL and by performing serial dilutions as detailed in Elkins et al. (2019) [85]. Standard blocks were periodically run throughout the analysis for quality assurance to ensure data integrity.

MS data were acquired on a Thermo Scientific Vanquish ultrahigh-performance liquid chromatography (UHPLC) system (Thermo Fisher Scientific, Waltham, MA, USA), coupled with a Q Exactive Plus mass spectrometer (Thermo Fisher Scientific, Waltham, MA, USA). The QE MS was set to positive electrospray ionisation (ESI) mode with a mass range of 80–1200 *m/z*, resolution at 35,000; normalised collision energy was at 30 V, and maximum ion time was 200 msec. The heated capillary temperature and the source temperature were maintained at 310 °C and 300 °C, respectively. The auxiliary, sheath and sweep gases (N_2_) were 15, 28 and 4 units, respectively. Spray voltage was set at 3.6 kV. The system was calibrated with Pierce LTQ Velos ESI Positive Ion Calibration Solution (Thermo Scientific, product no. 88323) prior to analysis.

Analytes were separated on a Phenomenex Luna Omega C_18_ 150 × 2.1 mm × 1.6 μm column with an injection volume of 3 µL. Separation was achieved as described in Elkins et al. (2019) [85] by using a multistep gradient starting at 60% solvent A (water with 0.1% formic acid) and 40% solvent B (acetonitrile with 0.1% formic acid). The gradient programme was performed as follows: 0–2 min 40% B, 2–3 min 40–75% B, 3–10 min 75–90% B, 10–11 min 90–100% B, 11–15 min 100% B, followed by equilibration to initial conditions for 5 min at a flow rate of 0.3 mL/min. The CBDA and THCA quantitation was performed by using UV DAD set to 280 nm, and the column was maintained at 30 °C as described in Elkins et al. (2019) [85]. Due to its better precision under certain ranges of cannabinoid concentrations, the UV system was utilised to detect THCA and CBDA values in samples where these cannabinoids’ concentration reads exceeded 2 mg/g. All mobile phase solvents were HPLC grade and were obtained from Fisher Scientific (Fair Lawn, NJ, USA). Acquired data were quantitatively processed using Tracefinder 5.1 build 110 (Thermo Fisher Scientific, Waltham, MA, USA).

To simplify the chemotypic profile assessment and to reflect the plant’s genetic potential for synthesising specific compounds, the quantity of each cannabinoid in its acidic form (e.g., CBDA) was converted to its compatible neutral form (e.g., CBD) through molecular weight transformations. The quantities of the converted and the measured neutral forms were summed to reflect the overall production of specific cannabinoids. Because the transformation of THC into CBN naturally accrues with the ageing of plant material or with its exposure to light and heat, in specific cases mentioned in this paper, the same method was used to convert CBN to THC in order to reflect the genuine quantities of the overall synthesised THC.

### 4.4. Statistical Analysis and Spatial Adjustments

Data analysis for the current study was performed using IBM SPSS Statistics for Windows, ver. 26.0 (Armonk, NY, USA: IBM Corp) and by R (R Core Team, 2020). Phenotypes were adjusted for spatial variability with autocorrelation error (AR1 × AR1) models to generate best linear unbiased estimates (BLUEs) using ASReml-R ver. 3 [86], where genotypes were fitted as a random effect. For CBC, CBN, TCC and THC, the column was fitted as an additional random effect in the model (Appendix A). Broad-sense heritability (H^2^) for each trait was calculated via ASReml models and represents the proportion of the total phenotypic variation (V_P_) that is attributable to the total genetic variation (V_G_) as H^2^ = V_G_/V_P_.

## 5. Conclusions

The present study provided evidence of the complex interplay between plant features, plant inflorescence morphology and a plant’s chemotypic profile. Notably, strong correlations were identified between vigorous growth rate during the vegetative phase, high-stature plants and wide inflorescences relating to the prolific production of cannabinoids. Additionally, the current study has expanded the research field by identifying that within genotypes, not only THC and CBD but also CBC, CBG, THCV and CBDV maintain steady qualitative traits and variable quantitative traits. Finally, built on these results, a successful model for the prediction of cannabinoid production was generated. These findings will have a significant impact on the breeding and cultivation of the chemotypically stable and reproducible cannabis genotypes that will facilitate the production of novel medicinal applications.

## Figures and Tables

**Figure 1 plants-12-00493-f001:**
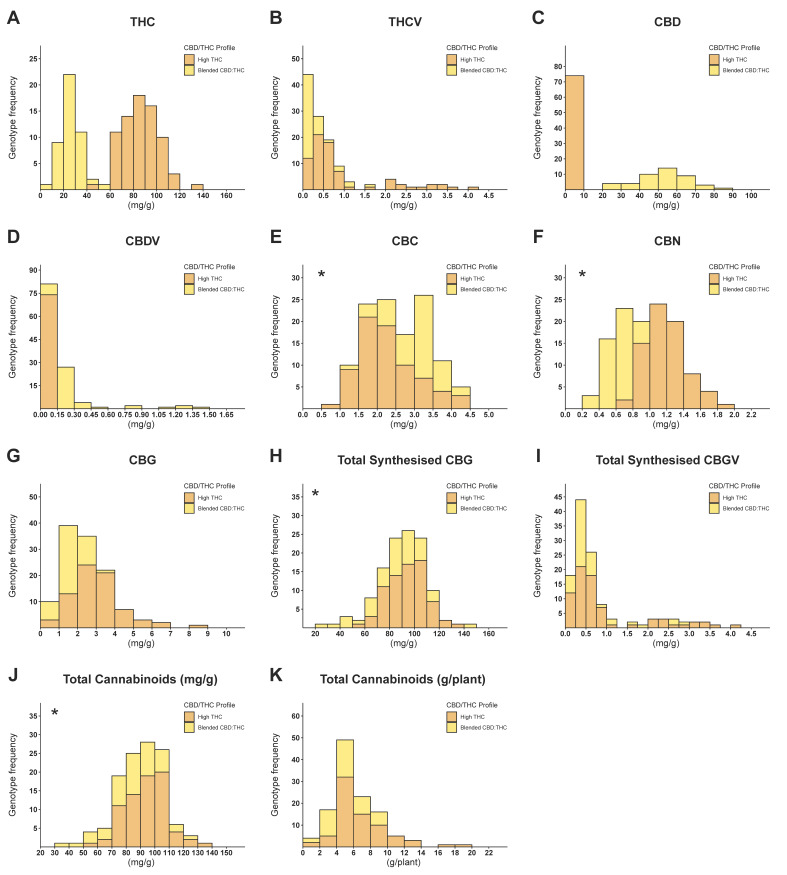
Frequency distribution of 119 cannabis genotypes across 11 cannabinoid parameters. Orange segments represent high-THC genotypes, and yellow segments represent blended THC:CBD genotypes. Charts marked with an asterisk on the top left corner are normally distributed. (**A**) Δ^9^-tetrahydrocannabinol (THC) content (mg/g), (**B**) Δ^9^-tetrahydrocannabivarin (THCV) content (mg/g), (**C**) cannabidiol (CBD) content (mg/g), (**D**) cannabidivarin (CBDV) content (mg/g), (**E**) cannabichromene (CBC) content (mg/g), (**F**) cannabinol (CBN) content (mg/g), (**G**) cannabigerol (CBG) content (mg/g), (**H**) total synthesised CBG (mg/g), (**I**) total synthesised CBGV (mg/g), (**J**) total cannabinoid concentration (mg/g), (**K**) total cannabinoid per plant (g/plant).

**Figure 2 plants-12-00493-f002:**
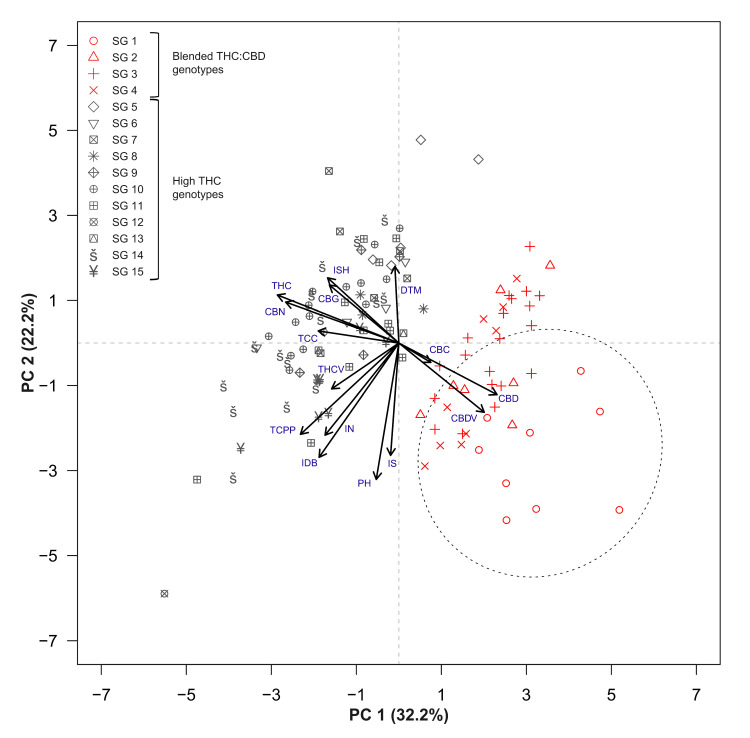
Principal component analysis (PCA) for the association between cannabinoid content, plants phenology, inflorescences morphology and key physiological attributes across 119 drug-type cannabis genotypes. Red symbols and black symbols indicate blended THC:CBD and high-THC genotypes, respectively. Symbol shapes specify the genotypic vernacular classification into strain groups (SG, such as “Northern Lights”, “Purple Kush” and “Sour Diesel”). The ellipse marked over the biplot’s 1st and 4th quadrants represents a 95% confidence level for genotypes classified to SG1 (red circle symbols), a group comprising accessions obtained from crossing a blended CBD:THC drug-type plant and an off-type hemp plant. Abbreviations: total cannabinoids per plant (TCPP).

**Figure 3 plants-12-00493-f003:**
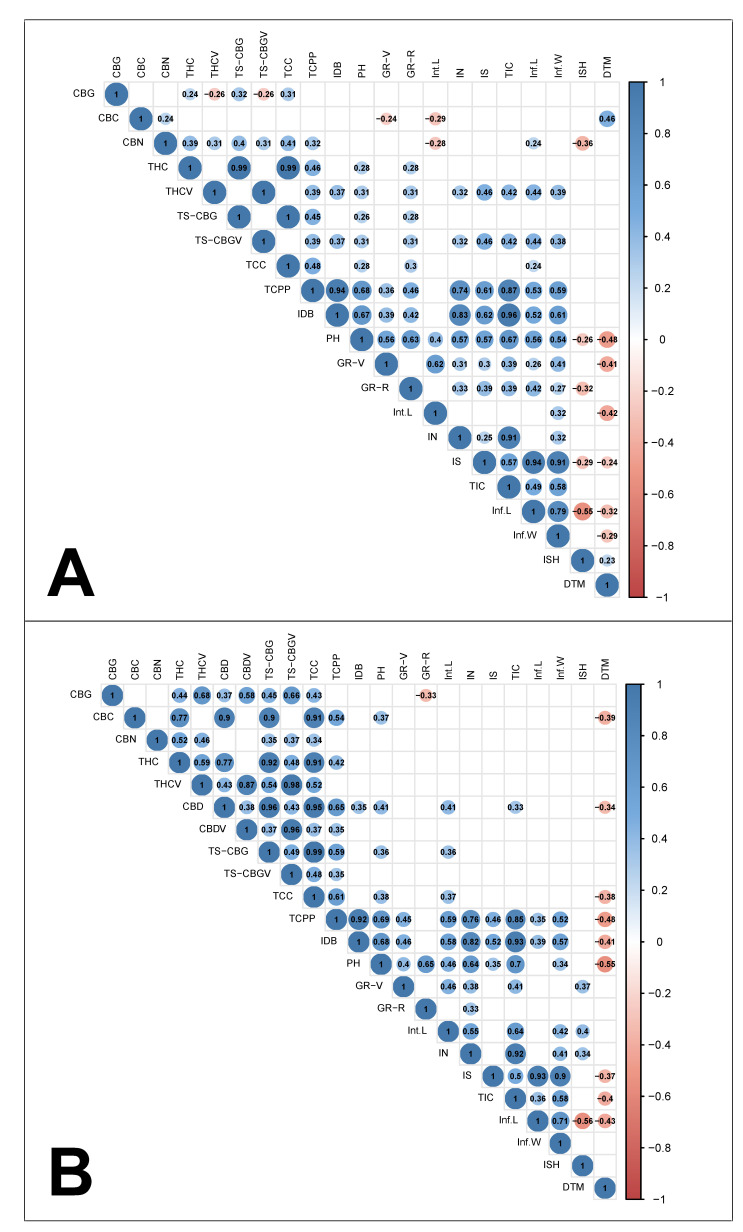
Correlation coefficient matrix for cannabinoid content, plant phenology, inflorescence morphology and key physiological attributes. (**A**) Correlations across high-THC genotypes (*n* = 74), (**B**) correlations across blended THC:CBD genotypes (*n* = 36). Values and colours indicate correlation strength and direction, respectively. Blank cells indicate insignificant correlations (*p* > 0.05). Abbreviations: total synthesised CBG (TS-CBG), total synthesised CBGV (TS-CBGV), growth rate during the vegetative phase (GR-V), growth rate during the reproductive phase (GR-R), total inflorescence coverage (TIC).

**Figure 4 plants-12-00493-f004:**
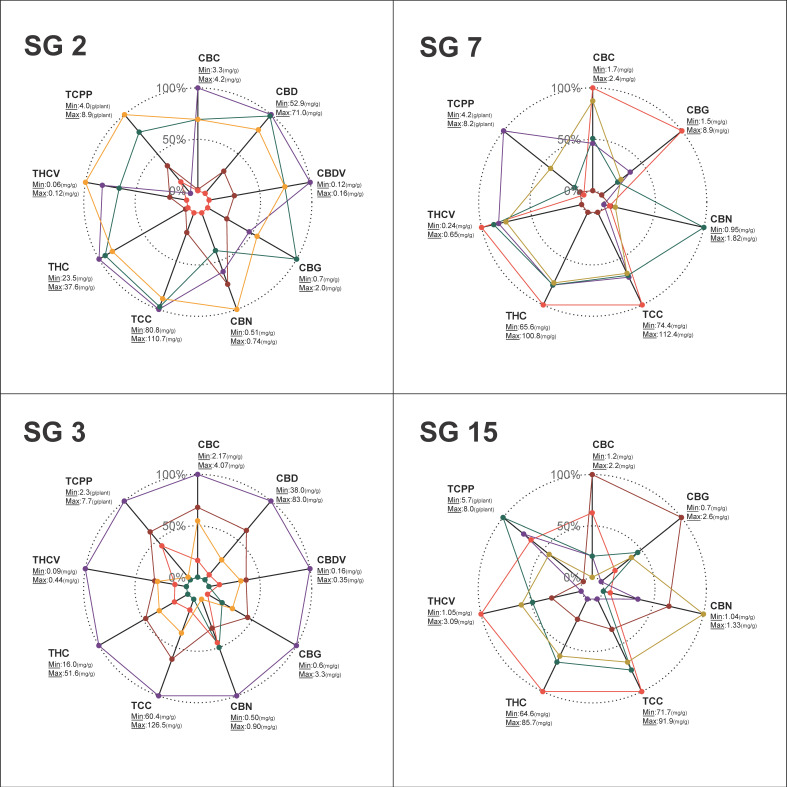
Radar charts for the relative quantities of cannabinoids across 4 vernacularly classified strain groups. Strain Group 2 (SG 2), Durga Mata (blended CBD:THC); Strain Group 3 (SG 3), Nebula (blended CBD:THC); Strain Group 7 (SG 7), Ice Cream (high THC); Strain Group 15 (SG 15), White Berry (high THC). Colours within each strain group indicate the genotypic replicate’s mean value for each parameter. The absolute range of the cannabinoid quantities, minimum (0%) and maximum (100%), across genotypes within each strain group is depicted next to the parameter indication. Abbreviations: strain group (SG).

**Figure 5 plants-12-00493-f005:**
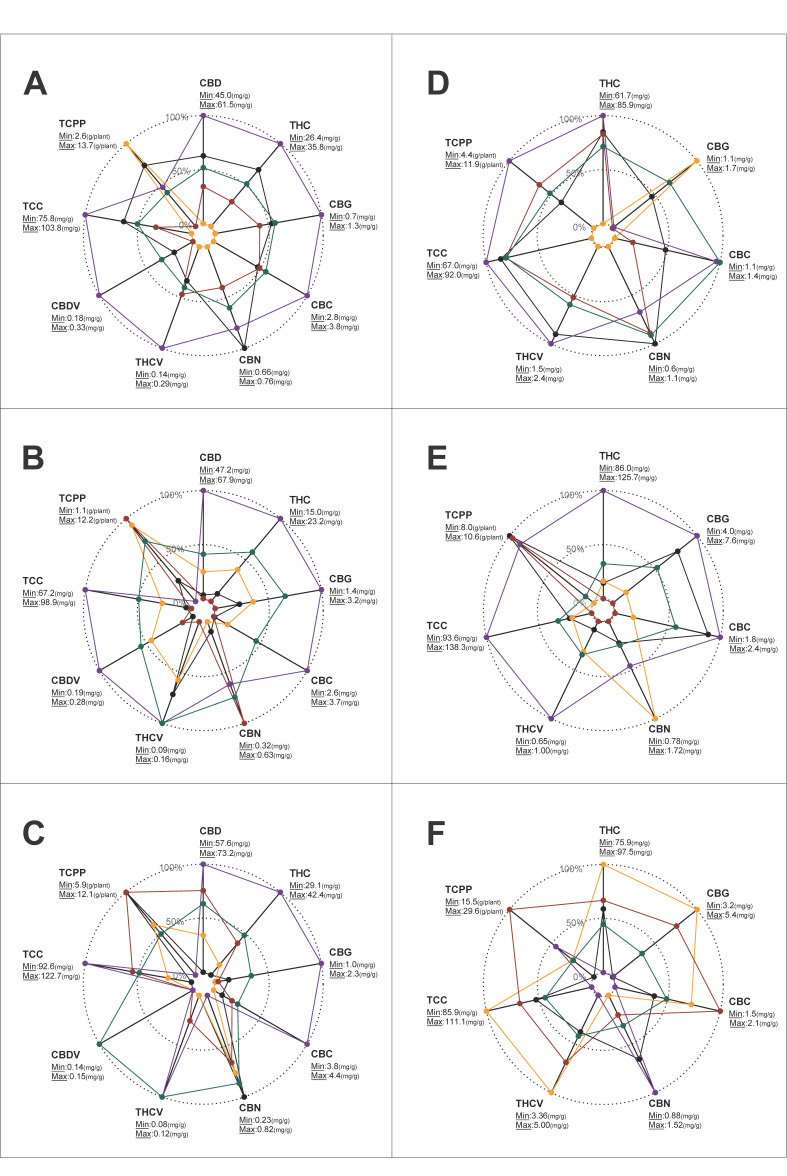
Radar charts for the relative quantities of cannabinoids recorded for 5 replicates (marked by colours) of 6 genotypes (**A**–**F**). The absolute range of cannabinoid quantities, minimum (0%) and maximum (100%), is presented next to each of the recorded parameters.

**Figure 6 plants-12-00493-f006:**
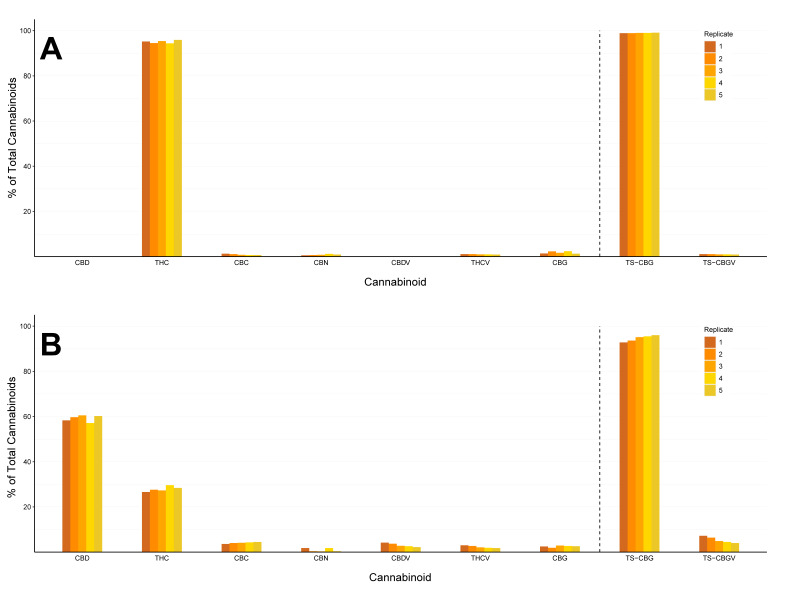
Variations in cannabinoid ratios (% of total cannabinoids) across 5 clonal plant replications. (**A**) High-THC genotype, (**B**) blended THC:CBD genotype. Experimental replicates are indicated by shades of colour.

**Table 1 plants-12-00493-t001:** Broad-sense heritability of cannabinoid biosynthesis, plant phenology, physiological traits and inflorescence morphology. Abbreviations: total cannabinoid concentration (TCC), days to maturation (DTM), plant height (PH), inflorescence dry biomass (IDB), average internode length (Int.L), inflorescence shape (ISH), inflorescence number (IN), average inflorescence size (IS), inflorescence width (Inf.W), inflorescence length (Inf.L).

Category	Trait	H^2^
Cannabinoids	THC	0.89
	THCV	0.95
	CBD	0.96
	CBDV	0.89
	CBC	0.79
	CBG	0.72
	CBN	0.51
	TCC	0.62
Phenology	DTM	0.49
Plant’s morphology	PH	0.52
	IDB	0.33
	Int.L	0.35
Inflorescences	ISH	0.38
	IN	0.29
	IS	0.15
	Inf.W	0.19
	Inf.L	0.18

**Table 2 plants-12-00493-t002:** Ratios between the highest and the lowest cannabinoid concentrations within each strain group. Strain group indices correspond with those presented in Figure 2 and Figure 4.

Strain Group		CBC	CBD	CBDV	CBG	CBN	THC	THCV	TCC	TCPP
2	Blended THC:CBD	1.3	1.3	1.3	3.0	1.5	1.6	2.0	1.4	2.2
3		1.9	2.2	2.2	5.7	1.8	3.2	4.9	2.1	3.3
7	High THC	1.4			5.9	1.9	1.5	2.7	1.5	1.9
15		1.9			3.9	1.3	1.3	2.9	1.3	1.4
Mean Ratio		1.6	1.8	1.8	4.6	1.6	1.9	3.1	1.6	2.2

**Table 3 plants-12-00493-t003:** The stability of the ratios between cannabinoid quantities (% of the overall TCC). Genotypes 446060 and 446039 correspond with those presented in Figure 6A and 6B, respectively. The presented THC quantity also comprises the produced CBN. Cannabidiol (CBD) and Δ^9^-tetrahydrocannabinol (THC) values include both the acidic and decarboxylated molecule forms. All cannabinoid transformations and amalgamates were performed through molecular weight conversion. The depicted σ represents a sample standard deviation (SD-S).

Genotype		CBD	THC	CBG	CBC	THCV	CBDV
ID:446060 (High-THC)	Proportions’ Mean	---	95.07%	1.81%	0.94%	1.02%	---
	σ	---	0.62%	0.52%	0.24%	0.07%	---
ID:446039 (Blended THC:CBD)	Proportions’ Mean	59.16%	27.90%	2.51%	4.08%	2.32%	3.09%
	σ	1.41%	1.16%	0.38%	0.35%	0.54%	0.81%

**Table 4 plants-12-00493-t004:** Multiple regression analysis for the prediction of total cannabinoids per plant (TCPP). (A) Regression coefficients and standard errors for high-THC genotypes (*n* = 74); (B) regression coefficients and standard errors for blended THC:CBD genotypes (*n* = 36).

TCPP	B	95% CI for B		SE B	β	R^2^	ΔR^2^
		LL	UL				
(A)							
**Model**						0.67	0.65 ***
**Constant**	−21.64 ***	−31.96	−11.32	5.17			
**PH**	0.13 ***	0.08	0.19	0.03	0.52		
**Int.L**	−1.11 *	−2.06	−0.16	0.48	−0.18		
**DTM**	0.19 *	0.05	0.34	0.07	0.22		
**Inf.W_10_**	2.87 ***	1.74	4.00	0.57	0.49		
(B)							
**Model**						0.66	0.63 ***
**Constant**	−13.68 ***	−19.91	−7.45	3.06			
**PH**	0.71 **	0.02	0.12	0.24	0.38		
**Int.L**	1.13	0.14	−0.38	0.74	0.2		
**Inf.W_10_**	1.77 **	0.55	2.98	0.6	0.4		

B = unstandardised regression coefficient, CI = confidence interval, LL = lower limit, UP = upper limit, SE B = standard error of the coefficient, β = standardised coefficient, R^2^ = coefficient of determination, ΔR^2^ = adjusted R^2^. * Significant at *p* < 0.05 level, ** significant at *p* < 0.01 level, *** significant at *p* < 0.001 level.

## Data Availability

The data presented in this study are available on reasonable request from the corresponding author.

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
