# Peer review of "The Cannabis Plant as a Complex System: Interrelationships between Cannabinoid Compositions, Morphological, Physiological and Phenological Traits"

_plants, 2023, doi:10.3390/plants12030493_

Round 1
Reviewer 1 Report
Journal: Plants (ISSN 2223-7747)
Manuscript ID: plants-2126334
Type: Article
Title: The cannabis plant as a complex system: Interrelationships between cannabinoid compositions, morphological, physiological and phenological traits
The present manuscript topic is investigated in the literature, and there is a very few of reference published. However, this paper gives significant contribution to the current knowledge in related field. The data are sound and it deserves to be published, after minor revisions.
?? Page# 1, Line # 30 & 31: THC, CBD, CBC, CBG, 31 THCV and CBDV stands for ??
Abbreviations must be described completely at first mention with brackets.
?? Keywords should not be the same as mentioned in the title or abstract.
?? Kindly don’t start a sentence with an abbreviation.
?? The text has many typing and grammatical errors, capitalization issues.
?? English style and language requires a profound revision. However, the readability of the manuscript needs to be improved, preferably carefully reviewing by a native English speaker.
?? Materials and methods must be at proper place.
?? It has been observed that the manuscript is submitted in speedy way, without reading the instructions completely.
?? The materials and methods section is very brief. Please add details for analytical methodologies to make it reproducible.
?? Quality assurance of data is mandatory!!! How many batch, repeats, chemical grade and for used instruments manufacturers’ user manual and instructions were strictly followed or not!!!
?? Collected data is sound one. It deserves to be published after minor improvements.
?? Use www.turnitin.com to find and eliminate unnecessary self-repetition and any copied text.
?? Very Minute Scientific Discussion.
?? Please cite Figure No. or Table No. in brackets at suitable places for a better connectivity in results and discussion sections as to facilitate the reader.
?? I would have expected slightly greater discussion; more detail on the mechanisms and logical reasoning is required. There is much more scope here for discussing the implications of what the results means.
?? Novelty of this research work is again questionable with reference to practical significance and economic feasibility must be worked and mentioned.
?? A few very old references have been used. These must be updated with recent research findings or removed.
?? Proper formatting is questionable. It must be according to MDPI Plants Journal. References formatting are inconsistent. A few DOI missing.
?? Verify each reference from original source and cross check references in the text and reference section.
Author Response
We would like to thank reviewer 1 for taking the time for assessing this manuscript. Your quality comments are very much appreciated and provided a clear way for the delivery of the scientific findings presented in this paper. Please find our responses to each of your comments below.
- Page# 1, Line # 30 & 31: THC, CBD, CBC, CBG, 31 THCV and CBDV stands for ?? Abbreviations must be described completely at first mention with brackets.
Thank you for pointing this out. We added a description for each of the abbreviations above (lines 31-33).
- Keywords should not be the same as mentioned in the title or abstract.
Thank you for bringing this to our attention. We removed all duplicated keywords that appeared in the manuscript title and/or abstract (lines 21-22).
- Kindly don’t start a sentence with an abbreviation.
We agree with this and fixed all sentences starting with an abbreviation throughout the manuscript (lines 295-296, 346-347).
- The text has many typing and grammatical errors, capitalization issues.
We found inconsistencies across the capitalization in the abbreviations used. The entire manuscript was edited to provide a systematic and uniform usage of acronyms.
- English style and language requires a profound revision. However, the readability of the manuscript needs to be improved, preferably carefully reviewing by a native English speaker.
According to your suggestion, the manuscript was reviewed by a native English speaker scientist. Edits were performed where needed throughout the entire manuscript (e.g., lines 61-65, 86, 93-94, 159, 376, 387, 435-436, 458,461, 467-468, 534-535).
- Materials and methods must be at proper place.
We relocated the material and methods section to the appropriate location (lines 712-722).
- It has been observed that the manuscript is submitted in speedy way, without reading the instructions completely.
We apologise for this. We checked the guidelines for authors provided on “Plants” website and made the following changes to the manuscript:
- Description for each abbreviation defined the first time they appear in the abstract, the main text and the first figure or table.
- The conclusion paragraph was removed and located after the material and methods section (lines 712-722).
- The materials and methods section is very brief. Please add details for analytical methodologies to make it reproducible.
We thank the reviewer for the suggestion, we have now reviewed and updated where appropriate in the "Cannabinoid quantification" section of the Materials and Methods. All required changes are clearly identified by track changes, and we believe this has improved the manuscript and will address any issues regarding the reproducibility of the methodology.
- Quality assurance of data is mandatory!!! How many batch, repeats, chemical grade and for used instruments manufacturers’ user manual and instructions were strictly followed or not!!!
All data were analysed in one batch with standards run periodically throughout the sequence for QC purposes. We have updated the manuscript to include specific details on standard and solvent suppliers and quality as well as more detail on methods of preparation. The manuscript has also been updated to indicate that standards were run at defined intervals throughout the sequence ensuring data integrity (these revisions occurred in the “Cannabinoid quantification section” in the Materials and Methods).
- Collected data is sound one. It deserves to be published after minor improvements.
Thank you.
- Use www.turnitin.com to find and eliminate unnecessary self-repetition and any copied text.
We used iThenticate to identify repeated/copied text. As noted, the software recognised some repetition in the Material and Methods section (originated from previous papers of the authors). We have addressed this issue by rewording the sentences where possible to substantially minimize the repetition with our previously published manuscript (these revisions occurred across lines 661-688 of the Materials and Methods section).
- Very Minute Scientific Discussion.
In order to enrich the scientific aspects of the manuscript, we expanded on some of the points presented in the discussion (please refer to comment 14).
- Please cite Figure No. or Table No. in brackets at suitable places for a better connectivity in results and discussion sections as to facilitate the reader.
We agree. Numbers of relevant Figures and Tables were inserted throughout the entire discussion (lines 401, 424, 448, 457, 459, 466, 471, 473, 475, 504, 514).
- I would have expected slightly greater discussion; more detail on the mechanisms and logical reasoning is required. There is much more scope here for discussing the implications of what the results means.
We added a section to the discussion providing more details regarding the relationship between the plant's morphology and a prolific synthesis of cannabinoids (lines 548-552).
- Novelty of this research work is again questionable with reference to practical significance and economic feasibility must be worked and mentioned.
To strengthen the practical aspects of the findings presented in this manuscript, we added sections to the discussion to exemplify the economic benefits behind the proposed selection methodology (lines 534-540).
- A few very old references have been used. These must be updated with recent research findings or removed.
We removed old references that could be replaced by more recent studies (numbers in brackets refer to the reference list in the original draft):
[1] Abel et al. 1980
[2] Zlas et al. 1993
[3] Russo et al. 2008
[16] Hammond et al. 1973
[21] de Meijer et al. 1992
[24] Haney et al. 1973
[28] Levin et al. 1973
[35] Sallan et al. 1975
- Proper formatting is questionable. It must be according to MDPI Plants Journal. References formatting are inconsistent. A few DOI missing.
We checked and fixed all formatting issues in the references. Missing DOIs were added where applicable.
- Verify each reference from original source and cross check references in the text and reference section.
The references used in this study were rigorously checked. All references in the body text were matched with the reference list.
Reviewer 2 Report
In this manuscript, Naim-Feil et al. investigated the interplays between traits associated with the intact plant (e.g., yield, phenology, growth rate), inflorescences morphology, and cannabinoid composition. Although the manuscript is attractive, there are some concerns that should be addressed.
-Generally, the manuscript is well organized but there are some typographical and grammatical errors.
-The paper title is well stated, it is informative and concise.
-Abstract is well structured.
-The introduction was not well written, and it is too briefly presenting the subject and research problem.
L43: "multipurpose source of recreational, ritual and medicinal applications" should be changed to "multipurpose source of industrial (10.3906/bot-1907-15), ornamental (https://doi.org/10.3390/plants11182383), and pharmaceutical (https://doi.org/10.1007/978-981-16-8822-5_4) applications".
L51: Provide reference (s): (https://doi.org/10.1016/j.biotechadv.2022.108074)
L57: Provide new reference (s): (https://doi.org/10.1016/j.isci.2021.103391; https://doi.org/10.1016/j.indcrop.2020.113026; https://doi.org/10.1111/nph.17140)
-Material and research methods are presented appropriately. The experimental setup and the description in the methods section are well structured, and the statistical analysis is correctly performed.
-The results obtained in this study are interesting. Results are presented correctly.
-In general, the discussion was not well written. This part should be improved.
Author Response
The authors would like to thank reviewer 2 for dedicating the time to provide insightful comments. Care has been taken to implement the required changes to improve the manuscript. Please find our responses to each of your comments below.
- The introduction was not well written, and it is too briefly presenting the subject and research problem.
Thank you for highlighting this issue. We edited the manuscript’s introduction and added more details regarding recent scientific studies that have examined the effect of environmental conditions on cannabinoid productivity (lines 104-117).
- L43: "multipurpose source of recreational, ritual and medicinal applications" should be changed to "multipurpose source of industrial (10.3906/bot-1907-15), ornamental (https://doi.org/10.3390/plants11182383), and pharmaceutical (https://doi.org/10.1007/978-981-16-8822-5_4) applications".
Thank you for this helpful suggestion. We implemented the recommended changes. However, we omitted the manuscript referring to the ornamental use of cannabis as this study presents a new approach/concept and therefore, it is not quite suitable for this sentence [For millennia, Cannabis sativa L. (cannabis) was extensively used by mankind….”, (lines 45-47)].
- L51: Provide reference (s): (https://doi.org/10.1016/j.biotechadv.2022.108074)
The reference has been added to the Introduction section (line 55)
- L57: Provide new reference (s): (https://doi.org/10.1016/j.isci.2021.103391; https://doi.org/10.1016/j.indcrop.2020.113026; https://doi.org/10.1111/nph.17140)
The references have been added to the Introduction section (line 61)
- Material and research methods are presented appropriately. The experimental setup and the description in the methods section are well structured, and the statistical analysis is correctly performed.
Thank you
- The results obtained in this study are interesting. Results are presented correctly.
Thank you
- In general, the discussion was not well written. This part should be improved.
To address this feedback, we carefully read the manuscript's discussion and identified some issues which required improvement. Most notably, we improved our discussion regarding the association between the plant's morphology and biology as well as practical applications of the findings presented in the study (an issue also raised by reviewer 1). More specifically, we added a description in the discussion section to clarify how the presented findings contribute significantly to practical aspects of cannabis breeding and cultivation (lines 538-540) as well as provide possible explanations for the association between morphological attributes and cannabinoids content (lines 548-552).
Round 2
Reviewer 2 Report
All the comments have been addressed. I think that the current form of the MS can be published in Plants.